

# STEP: A tool to perform tests of smoothness on differential distributions based on expansion of polynomials

**Patrick L. S. Connor[1]⋆ and Radek Žlebčík[2]**

**1** Institut für Experimentalphysik & Center for Data and Computing in natural Sciences,
Universität Hamburg, Hamburg, Germany
**2** Charles University, Prague, Czech Republic

⋆ patrick.connor@desy.de

## Abstract

We motivate and describe a method based on fits with polynomials to test the smoothness of differential distributions. As a demonstration, we apply the method to several measurements of inclusive jet double-differential cross section in the jet transverse momentum and rapidity at the Tevatron and LHC. This method opens new possibilities to test the quality of differential distributions used for the extraction of physics quantities such as the strong coupling.

## 1 Introduction

The limited resolution of detectors, the reconstruction algorithms, the analysis techniques, certain approximations, or a combination of all these may cause artificial deviations of experimental distributions from an expected smooth behaviour. An example is the inclusive jet

differential cross section at hadronic colliders, such as the Tevatron and LHC [1,2], where the data points play a crucial role in the extraction of the strong coupling [3] and of the parton distribution functions (PDFs) [4]. These measurements cover a large phase space and the differential cross sections span over several orders of magnitude, with a statistical precision of the level of one percent. In practice, it has been reported that the inclusive jet measurements with data from LHC Run 1 are difficult to include in the global PDF fits [5–7].

Typical sources for deviations from a smooth behaviour in such a spectrum are:

- different triggers to obtain a spectrum over a large range of values (which may appear in the form of steps in the spectrum);

- calibrations as a function of the same variables as the observable (especially if the calibrations are provided with a coarser binning scheme than the one used for the observable, also resulting in steps);

- the neglecting of correlations between the bins of the spectrum (e.g. for multi-count observables, or for any observable after a procedure of unfolding [8], resulting for instance in the apparent movements of adjacent points in opposite directions).

Tests of smoothness performed at all stages of the analysis help assess the impact of every step of the data reduction, and estimate the quality of a spectrum before any use in global PDF fits. A compact description of distributions is also useful in the searches for New Physics, which are often expected to manifest as a bump on top of a smooth Standard Model background [9]. Alternatively, a smooth fit of a spectrum may be useful as a smoothing procedure, e.g. to estimate smooth systematic variations when the original estimate suffers from statistical fluctuations due to the limited statistics of the simulation in an unfolding procedure.

In this article, we present an iterative method to perform a smooth fit of a spectrum with a large number of points, based on expansion of polynomials, and present several applications in the context of inclusive jet production in hadronic collisions [10–14]. We discuss the determination of the optimal order of the polynomial with various stopping criteria. We also provide an implementation of the test so that it may be applied to other observables, possibly beyond high energy physics.

## 2 Method

The method was primarily developed using Chebyshev polynomials of the first kind [15], which are defined iteratively as follows:

$$T_0(x) = 1, \tag{1}$$

$$T_1(x) = x, \tag{2}$$

$$T_{i+1}(x) = 2x T_i(x) - T_{i-1}(x), \tag{3}$$

with $x \in [-1, 1]$. The first polynomials are shown in Figure 1a. A spectrum $f$ can be approximated by a polynomial $f_n$ of order $n$:

$$f_n(x) = \sum_{i=0}^{n} b_i T_i(x). \tag{4}$$

The interpolation with such polynomials ensures that the original coefficients of $f_n$ stay similar when a term of higher order is added, which helps set up an iterative fit procedure. The method

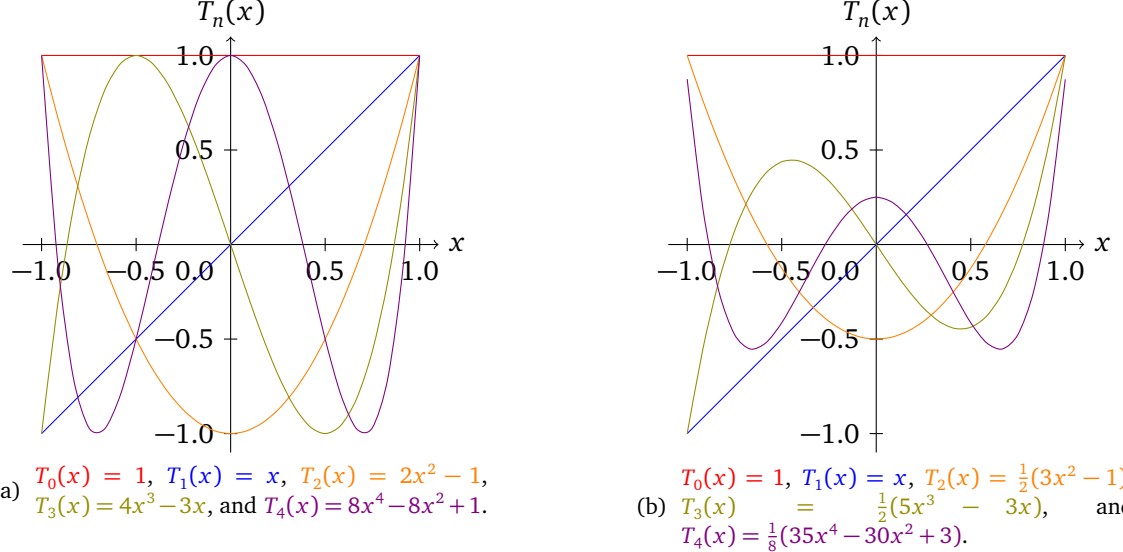

(a) $T_0(x) = 1$, $T_1(x) = x$, $T_2(x) = 2x^2 - 1$, $T_3(x) = 4x^3 - 3x$, and $T_4(x) = 8x^4 - 8x^2 + 1$.

(b) $T_0(x) = 1$, $T_1(x) = x$, $T_2(x) = \frac{1}{2}(3x^2 - 1)$, $T_3(x) = \frac{1}{2}(5x^3 - 3x)$, and $T_4(x) = \frac{1}{8}(35x^4 - 30x^2 + 3)$.

Figure 1: First orders of Chebyshev polynomials of first kind (left) and Legendre polynomials (right).

also works with other bases such as Legendre polynomials [16], shown in Figure 1b, with the same first two terms and the following recursive formula:

$$(i + 1)T_{i+1}(x) = (2i + 1)x\,T_i(x) - i\,T_{i-1}(x),\tag{5}$$

also with $x \in [-1, 1]$. However, for the present purpose, it is sufficient to find one basis that satisfies these properties.

In the case of inclusive jet cross section differential in the transverse momentum $p_T$, the range of the variable should be mapped to the range of the measurement; as it is steeply falling, we also take the logarithm of $p_T$ and the exponential of the expansion to avoid biasing the fit toward one part of the spectrum and have a numerically more stable result despite the different orders of magnitude:

$$f_n(p_T) = \exp\left(\sum_{i=0}^{n} b_i T_i\left(2\frac{\log p_T / \log p_T^{\min}}{\log p_T^{\max} / \log p_T^{\min}} - 1\right)\right),\tag{6}$$

where $n$ is the degree of the polynomial. To find the optimal value of $n$ to fit the spectrum within given uncertainties, the following objective function is defined:

$$\chi_n^2 = \min_{b_{i \leq n}}\left[\left(\mathbf{x} - \mathbf{y}_{b_{i\leq n}}\right)^{\mathsf{T}} \mathbf{V}^{-1}\left(\mathbf{x} - \mathbf{y}_{b_{i\leq n}}\right)\right],\tag{7}$$

where

- $\mathbf{x}$ corresponds to the binned differential distribution;

- $\mathbf{y}_{b_{i\leq n}}$ corresponds to the integral of $f_n$ normalised to the bin width for a given set of parameters $b_{i\leq n}$;

- $\mathbf{V}$ is the covariance matrix of $\mathbf{x}$ describing the uncertainties.

The iterative fit procedure is the following:

1. the two lowest-order coefficients $b_0$ and $b_1$ are obtained from the first and last points of the spectrum;

2. the next coefficient $b_n$ ($n = 2$ for the first iteration) is released and a fit is run with coefficients $b_0, b_1, \ldots, b_n$ free in the fit;

3. the previous step is iterated until a stopping criterion is satisfied.

Four flavours of the test are proposed, corresponding to four different stopping criteria, summarised in Table 1 and whose choice is left to the user:

- The degree of the polynomial has reached a predefined, maximal value.

- The $\chi_n^2$ divided by the number of degrees of freedom (ndf) has reached a plateau and is no longer decreasing, or is compatible with unity within $k$ standard deviations:

$$|\chi_n^2 - \text{ndf}| < k\sqrt{2\,\text{ndf}}. \tag{8}$$

- A F-test [17] is run to compare the fits with degrees $n$ and $n+1$. The iterative process is then interrupted if the obtained $p$-value is lower than a predefined threshold $p$.

- We apply cross-validation techniques [18] on statistical replicas of the original data set. First we arrange these replicas into set of $N$ pairs, each pair including a training and a validation replica. We compare the fits of degrees $n$ and $n+1$ performed on the training replica by evaluating the $\chi_n^2$ and $\chi_{n+1}^2$ on the corresponding validation replica. We stop if the $\chi^2$ improves, i.e. $\chi_{n+1}^2 < \chi_n^2$, for smaller fraction of replicas than a predefined threshold $p$.

In the present study, we investigate all stopping criteria, but the results shown in the various figures are all based on cross validation with $N = 10,000$ pairs of replicas (respectively for training and validation) and a threshold of 90% ($p = 0.9$).

The fit Ansatz is general and only assumes a smooth distribution. No other hypothesis is made on the nature of the distribution under scrutiny.

The tool, called STEP,[1] consists in a header-only file written in C++17 and relies only on ROOT and STL.[2] It is available on GitLab at CERN [20] under free license, along with its Doxygen documentation and several applications that will be described in the next section. The test of smoothness can be run using a simple call of the function `Step::GetSmoothFit()`, which takes templates arguments:

- the basis (by default Chebyshev polynomials)

- and possible options to rescale the input and output variables (logarithm and exponential in the case of inclusive jet spectrum, as shown in Eq. 6);

and normal arguments:

- a histogram in ROOT format,

- a covariance matrix (optional),

- the maximal degree allowed in the iterative procedure ($n$),

- an interval of bins of the histograms in which the fit should be performed (optional),

- options to control the stopping criterion (optional, see Table 1).

The results of each iteration from the last call of the function are stored in a global variable `Step::chi2s`. The Doxygen documentation may be found in the GitLab repository.

---

[1]*Smoothness Tests with Expansion of Polynomials.*

[2]During the review of the present paper, an independent version of the algorithm written in Python has also been provided [19].

Table 1: Stopping criteria, with options in the tool and parameter of the method.

| method | option | parameter |
|---|---|---|
| none | `None` | degree of polynomial ($n$). |
| Eq. 8 | `Chi2ndf` | number of standard deviations. |
| F-test | `fTest` | threshold for $p$-value. |
| Cross validation | `xVal` | threshold for fraction of validation replicas with better $\chi^2$/ndf. |

# 3 Applications

We illustrate the method with different cases in the context of inclusive jet production measured at the Tevatron and LHC used in global PDF fits [10–14]. All examples are available on the GitLab repository of the tool. The measurements are investigated in the chronological order of publication.

## 3.1 Measurements at Tevatron with $\sqrt{s} = 1.96\,\text{TeV}$

The CDF and DØ Collaborations have provided measurements of double-differential inclusive jet cross sections in proton-antiproton collisions at a centre-of-mass energy $\sqrt{s} = 1.96\,\text{TeV}$ [21, 22]; jets are clustered using the midpoint algorithms with a distance parameter $R = 0.7$ [23]. No description of the bin-to-bin statistical correlations is provided by any collaboration. The DØ measurement includes an additional bin-to-bin uncorrelated systematic uncertainty at the percent level.

Tests of smoothness are performed in each rapidity $y$ bin separately on the CDF (DØ) measurement, accounting for statistical uncertainties only (both statistical and bin-to-bin uncorrelated systematic uncertainties). The $\chi^2$/ndf and fit probabilities are shown in Table 2 (Table 4) for polynomials of various degrees up to $n = 5$ ($n = 6$). In the upper panel of Figure 2 (Figure 4), we also show the ratio to another fit Ansatz inspired from Ref. [9], labelled Harris and Kousouris (HK):

$$\text{HK}(p_\text{T}) = p_0 \frac{\left(1 - \frac{p_\text{T}}{p_4}\right)^{p_2}}{p_\text{T}^{p_1 + p_3 \log(p_\text{T})}} \,. \tag{9}$$

In its original form (i.e. dijet mass instead of $p_\text{T}$, with $p_3 \equiv 0$, and with $p_4 \equiv \sqrt{s}$), this function is inspired from physics expectations in the context of dijet mass cross sections; the $p_3$ parameter is introduced to cover higher orders, and the $p_4$ parameter let free, to adapt to the inclusive jet $p_\text{T}$ spectrum. For the same number of parameters (three last $y$ bins of each measurement; second and three last $y$ bins of DØ measurement), we can directly compare the Step and HK fits: they do show a similar fit performance, although Step fits have converged more easily and required less tuning of the initial parameters and parameter ranges. In the lower panel of the same figures, the pulls (i.e. normalised residuals) are shown for both fits, to help identify outliers. It also allows to see certain bins that were not visible in the vertical axis range of the upper panel.

The results from the F-test and cross validation are shown in Table 3 (Table 5). Figure 3 (Figure 5) shows the average $\chi^2$s obtained from the validation replicas. Finally, an Asimov data set [24] is generated using PYTHIA 8.2 [25] Monte Carlo generator. This generated sample has much higher number of events than recorded at the Tevatron to ensure a smooth spectrum by construction; the binning and the uncertainties are by construction always taken from the respective data set. In each $y$ bin, the fit is repeated also on the Asimov data set; however, the same polynomial degree as obtained for the fit to data is forced (also shown in Figures 2-4).

In the first $y$ bin of the CDF measurement, Figure 2, where statistical uncertainties are the

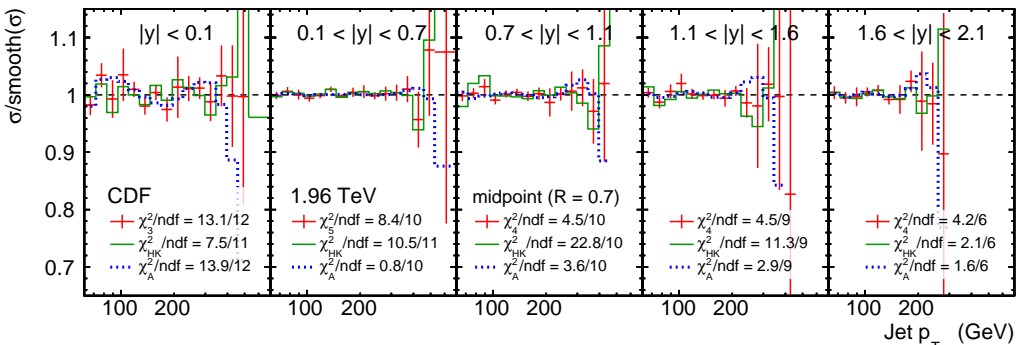

(a) The red points (green plain line) show(s) the ratio to the smooth fit with the expansion of polynomials (HK function); the blue dotted line shows the ratio of Asimov data fitted with a polynomial of the same degree as for the real data in the same $y$ bin. The error bars show the relative statistical uncertainties.

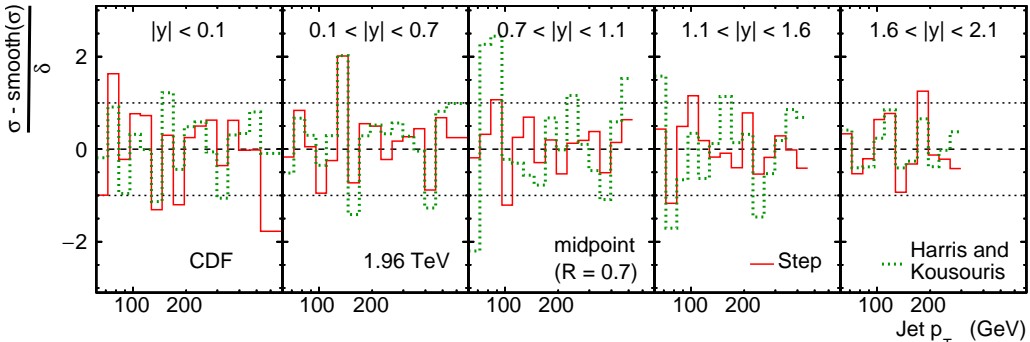

(b) Bin-by-bin pulls, shown with the same convention. The horizontal, dotted lines correspond to one standard deviation.

Figure 2: Tests of smoothness in each $y$ bin of CDF measurement [21]. In the $y$ axis title, $\sigma$ (smooth($\sigma$)) stands for cross section (smooth fit of the cross section); $\delta$ indicates the total bin-to-bin uncorrelated uncertainties.

largest, the fit performance on the Asimov data and real data is the same, showing that no effect besides the statistical fluctuations is noticeable. In both cases, the fit is driven by the bins with smaller statistical uncertainties. The lower fit performance in the second $y$ bin seems related to the presence of steps in the $p_T$ spectrum at 146 GeV. Similar steps can be seen in the third and fourth $y$ bin at 96 GeV, but the statistical uncertainties are also much larger. These steps are also visible with the HK function, but not visible with the Asimov data set, and correspond to trigger thresholds (which is also confirmed by the decrease of the statistical uncertainties in the spectrum). In other words, we see here some strong indication for uncorrected trigger inefficiencies.

Such steps due to trigger inefficiencies can also be suspected in the three first $y$ bins of the DØ measurement with both the Step and HK fits, Figure 4: more visible on the pulls (lower panel), the degree of the polynomial is one unity higher and the $\chi^2/$ndf values of the Asimov data lower than in the three last $y$ bins. The F-test and the cross validation also indicate the need of higher orders in the second $y$ bin. Instead, in these three last $y$ bins, such steps are no longer noticeable, probably negligible in comparison with the larger bin-to-bin uncertainties, and none of the three stopping criteria indicate any need to increase the order.

Table 2: Fit performance in each $y$ bin of the CDF measurement [21].

(a) $\chi_n^2/\mathrm{ndf} \pm \sqrt{2\mathrm{ndf}}$

| $n$ | $\|y\| < 0.1$ | $0.1 < \|y\| < 0.7$ | $0.7 < \|y\| < 1.1$ | $1.1 < \|y\| < 1.6$ | $1.6 < \|y\| < 2.1$ |
|---|---|---|---|---|---|
| 1 | $237.97 \pm 0.38$ | $2791.94 \pm 0.38$ | $1482.33 \pm 0.39$ | $2012.18 \pm 0.41$ | $1729.24 \pm 0.47$ |
| 2 | $11.09 \pm 0.39$ | $131.61 \pm 0.39$ | $78.41 \pm 0.41$ | $77.92 \pm 0.43$ | $55.35 \pm 0.50$ |
| 3 | $1.09 \pm 0.41$ | $12.75 \pm 0.41$ | $10.81 \pm 0.43$ | $2.39 \pm 0.45$ | $2.69 \pm 0.53$ |
| 4 | $0.65 \pm 0.43$ | $1.61 \pm 0.43$ | $0.45 \pm 0.45$ | $0.50 \pm 0.47$ | $0.69 \pm 0.58$ |
| 5 | $0.71 \pm 0.45$ | $0.84 \pm 0.45$ | $0.48 \pm 0.47$ | $0.43 \pm 0.50$ | $0.26 \pm 0.63$ |

(b) Fit probabilities

| $n$ | $\|y\| < 0.1$ | $0.1 < \|y\| < 0.7$ | $0.7 < \|y\| < 1.1$ | $1.1 < \|y\| < 1.6$ | $1.6 < \|y\| < 2.1$ |
|---|---|---|---|---|---|
| 3 | 0.36 | 0.00 | 0.00 | 0.01 | 0.01 |
| 4 | 0.79 | 0.09 | 0.92 | 0.87 | 0.66 |
| 5 | 0.71 | 0.59 | 0.89 | 0.91 | 0.94 |

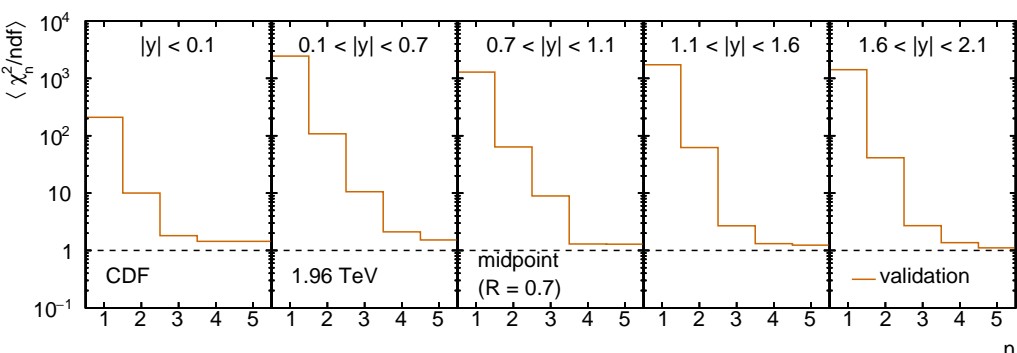

Figure 3: Average $\chi_n^2/\mathrm{ndf}$ from validation replicas and fits of validation replicas in each $y$ bin of the CDF measurement [21].

## 3.2 Measurements at LHC with $\sqrt{s} = 8\,\mathrm{TeV}$

The CMS and ATLAS Collaborations have provided measurements of double-differential inclusive jet cross sections in proton-proton collisions at a centre-of-mass energy $\sqrt{s} = 8\,\mathrm{TeV}$ [26, 27], where jets are clustered using the anti-$k_T$ algorithm with respective distance parameters $R = 0.7$ and $R = 0.6$ [28, 29].

The CMS measurement is provided with correlation tables for each $y$ bin, describing the correlations among the $p_T$ bins; it also includes a 1% bin-to-bin uncorrelated systematic uncertainty to account for small inefficiencies (e.g. trigger, jet identification). The ATLAS measurement is provided with ten thousand replicas that can be used to extract similar correlation tables with the bootstrap method [30]; additionally, among the systematic uncertainties, many result from the limited statistics of the Monte Carlo sample used in the data reduction, and may affect the shape of the spectrum only locally.

Tests of smoothness are performed in each $y$ bin separately on the CMS (ATLAS) measurement by accounting for both statistical and bin-to-bin uncorrelated systematic uncertainties (statistical uncertainties and systematic uncertainties related to the limited statistics of the numerical analysis). The $\chi^2/\mathrm{ndf}$ and fit probabilities are shown in Table 6 (Table 8) for polynomials of various degrees. The results from the F-test and cross validation are shown in Table 7 (Table 9), and the average $\chi^2/\mathrm{ndf}$ values for the validation replicas in Figure 7 (Figure 9). For CMS, the tests of smoothness in the six $y$ bins lead to $\chi^2/\mathrm{ndf}$ systematically lower than one. The low $\chi^2/\mathrm{ndf}$ values suggest the additional bin-to-bin systematic

Table 3: Comparing F-test and cross validation results for CDF measurement with different numbers of parameters.

(a) $|y| < 0.1$

F-test

Cross validation

|   | 3 | 4 |
|---|---|---|
| 4 | 0.9883 | |
| 5 | 0.9512 | 0.0204 |

|   | 3 | 4 |
|---|---|---|
| 4 | 0.8895 | |
| 5 | 0.8898 | 0.5009 |

(b) $0.1 < |y| < 0.7$

|   | 3 | 4 |
|---|---|---|
| 4 | 1 | |
| 5 | 1 | 0.9923 |

|   | 3 | 4 |
|---|---|---|
| 4 | 1 | |
| 5 | 1 | 0.9392 |

(c) $0.7 < |y| < 1.1$

|   | 3 | 4 |
|---|---|---|
| 4 | 1 | |
| 5 | 1 | 0.4107 |

|   | 3 | 4 |
|---|---|---|
| 4 | 1 | |
| 5 | 1 | 0.5715 |

(d) $1.1 < |y| < 1.6$

|   | 3 | 4 |
|---|---|---|
| 4 | 0.9998 | |
| 5 | 0.9996 | 0.8537 |

|   | 3 | 4 |
|---|---|---|
| 4 | 0.9868 | |
| 5 | 0.9879 | 0.6998 |

(e) $1.6 < |y| < 2.1$

|   | 3 | 4 |
|---|---|---|
| 4 | 0.9963 | |
| 5 | 0.9988 | 0.9795 |

|   | 3 | 4 |
|---|---|---|
| 4 | 0.9765 | |
| 5 | 0.9820 | 0.7998 |

uncertainty of 1% to be too conservative; Table 6a shows that given the bin-to-bin uncorrelated uncertainties, polynomials of degree 4 should in principle suffice. Instead, apart in the first $y$ bin, the cross validation has led to polynomials of degree 5. For ATLAS, the tests in the two first $y$ bins terminate with $\chi^2/\text{ndf}$ systematically larger than 1, and increasing the number of parameters does not show any improvement; in the four next $y$ bins, polynomials of degree 5 are found with $\chi^2/\text{ndf}$ compatible with unity. For both measurements, the F-test shows a similar tendency as the cross validation to increase the order in contrast to the criterion based on the $\chi^2/\text{ndf}$.

The tests obtained with cross validation are shown in upper panel of Figure 6 (Figure 8). As for the Tevatron experiment, certain bins may not be visible because of the vertical axis range, but these bins usually have large statistical uncertainties. Correlations matrices are also provided in the lower panel: the CMS measurement exhibits correlations and anti-correlations with a regular pattern, which is a known feature of the unfolding algorithm used in that analysis; the ATLAS measurements does not exhibit any anti-correlations, but regions centred at 90, 300, and 700 GeV show rather large correlations due to their in-situ calibration methods. With such correlations, it is more difficult to identify a clear step or an obvious outlier. Nevertheless, one can still observe steps possibly due to trigger inefficiencies in the CMS measurement, Figure 6, for instance at 507 GeV, especially in the three first $y$ bins. This value is explicitly mentioned as a trigger threshold in Ref. [26]. However, as they are outperformed in all $y$ bins by the Step fits, the HK fits are not shown; additional orders seem necessary to catch all effects in the shape of the spectrum.

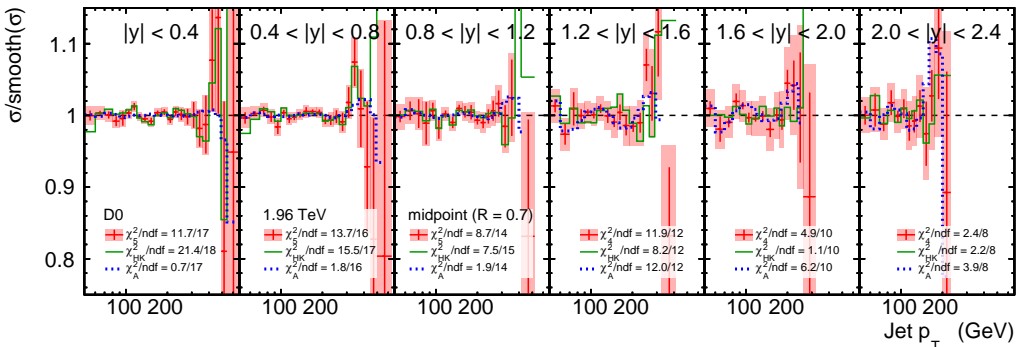

(a) The red points (green plain line) show(s) the ratio to the smooth fit with the expansion of polynomials (HK function); the blue dotted line shows the ratio of Asimov data fitted with a polynomial of the same degree as for the real data in the same $y$ bin. The vertical bars on the red points show the statistical contribution only, while the shaded areas show the statistical and systematic uncertainties related to the limited statistics of the numerical analysis added in quadrature.

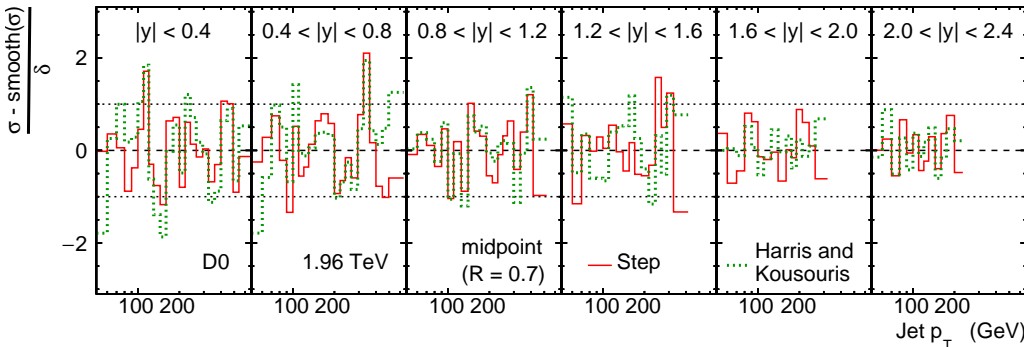

(b) Bin-by-bin pulls, shown with the same convention. The horizontal, dotted lines correspond to one standard deviation.

Figure 4: Tests of smoothness in each $y$ bin of DØ measurement [22]. In the $y$ axis title, $\sigma$ (smooth($\sigma$)) stands for cross section (smooth fit of the cross section); $\delta$ indicates the total bin-to-bin uncorrelated uncertainties.

The figures also include Asimov data sets produced with proton-proton collisions at 8 TeV, similarly as for the Tevatron measurements. In general, in all $y$ bins of both measurements, the low $\chi^2/\text{ndf}$ values indicate that a lower degree would be sufficient at describing the shape of the measurement within the given uncertainties and correlations. In fact, although this is not shown in the figures, polynomials of degree 4 would be sufficient in all $y$ bins.

Additional decorrelation may still arise from the combination of distinct systematic effects, even though these are bin-to-bin fully correlated. These were not included to run the Step fits, but may be considered (except the ones related to the luminosity) to compute a global $\chi^2/\text{ndf}$, also accounting for cross $y$ correlations. Doing so, one obtains global values of $\chi^2/\text{ndf} = 46.54/148 = 0.3144$ for CMS and $\chi^2/\text{ndf} = 123.8/135 = 0.9171$ for ATLAS.

Table 4: Fit performance in each $y$ bin of the DØ measurement [22].

(a) $\chi_n^2/\text{ndf} \pm \sqrt{2\text{ndf}}$

| $n$ | $\|y\| < 0.4$ | $0.4 < \|y\| < 0.8$ | $0.8 < \|y\| < 1.2$ | $1.2 < \|y\| < 1.6$ | $1.6 < \|y\| < 2.0$ | $2.0 < \|y\| < 2.4$ |
|---|---|---|---|---|---|---|
| 1 | $1185.74 \pm 0.31$ | $1687.53 \pm 0.32$ | $1473.09 \pm 0.33$ | $795.89 \pm 0.37$ | $618.56 \pm 0.39$ | $571.77 \pm 0.43$ |
| 2 | $70.88 \pm 0.32$ | $102.90 \pm 0.32$ | $99.75 \pm 0.34$ | $67.52 \pm 0.38$ | $47.81 \pm 0.41$ | $36.64 \pm 0.45$ |
| 3 | $8.89 \pm 0.32$ | $10.90 \pm 0.33$ | $8.43 \pm 0.35$ | $4.81 \pm 0.39$ | $4.48 \pm 0.43$ | $2.73 \pm 0.47$ |
| 4 | $1.02 \pm 0.33$ | $1.57 \pm 0.34$ | $1.05 \pm 0.37$ | $0.99 \pm 0.41$ | $0.49 \pm 0.45$ | $0.30 \pm 0.50$ |
| 5 | $0.69 \pm 0.34$ | $0.86 \pm 0.35$ | $0.62 \pm 0.38$ | $0.65 \pm 0.43$ | $0.14 \pm 0.47$ | $0.30 \pm 0.53$ |
| 6 | $0.73 \pm 0.35$ | $0.57 \pm 0.37$ | $0.54 \pm 0.39$ | $0.57 \pm 0.45$ | $0.14 \pm 0.50$ | $0.24 \pm 0.58$ |

(b) Fit probabilities

| $n$ | $\|y\| < 0.4$ | $0.4 < \|y\| < 0.8$ | $0.8 < \|y\| < 1.2$ | $1.2 < \|y\| < 1.6$ | $1.6 < \|y\| < 2.0$ | $2.0 < \|y\| < 2.4$ |
|---|---|---|---|---|---|---|
| 4 | 0.43 | 0.06 | 0.40 | 0.45 | 0.90 | 0.97 |
| 5 | 0.82 | 0.62 | 0.85 | 0.79 | 1.00 | 0.95 |
| 6 | 0.77 | 0.90 | 0.90 | 0.84 | 1.00 | 0.96 |

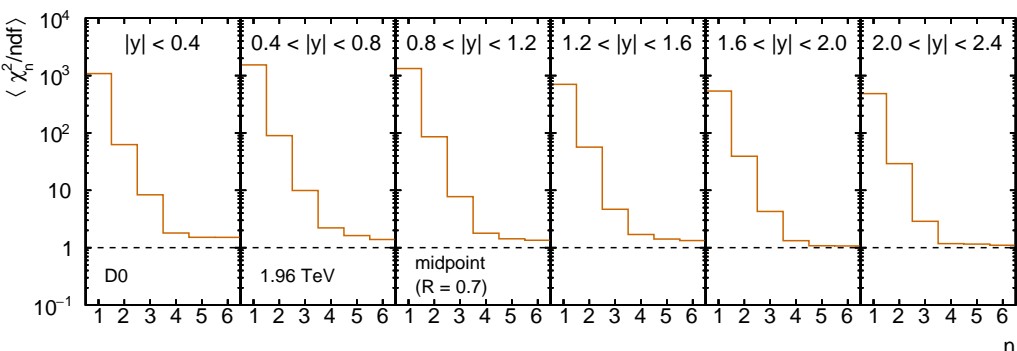

Figure 5: Average $\chi_n^2/\text{ndf}$ from validation replicas and fits of validation replicas in each $y$ bin of the DØ measurement [22].

### 3.3 Interpretation

For all $y$ bins of all four measurements, smooth fits with six or less parameters (i.e. $n \leq 5$) seem to be sufficient to describe the spectra within the given uncertainties and correlations. In general, it is remarkable to find that such a small number of parameters is sufficient to describe the shape of the inclusive jet spectrum over such a large range. This value may change for different stopping criteria, but usually not more than by one unity and only in certain $y$ bins.

The fits with Asimov data often indicate that less parameters are necessary to describe the shape within the given uncertainties and correlations: it likely indicates effects introduced in the data reduction, e.g. the combination of different triggers.

The HK function performs reasonably well for the Tevatron measurements, with a similar performance from the Step function; for the LHC measurements, however, the Step Ansatz significantly outperforms the HK one. One may include higher orders in Eq. 9, but this would return to a polynomial expansion. Furthermore, given the respective precisions of the measurements, the Asimov data show that five parameters are already sufficient to describe the shape of the spectrum; , adding more terms to the HK function seems unnecessarily complicated for these measurements.

To conclude, tests of smoothness may be useful to identify tensions before the data are published and handed over to global PDF collaborations. The respective measurements by ATLAS and CMS provide here two different examples. The set of systematic uncertainties provided with the ATLAS measurement covers the scattering of all points around a smooth behaviour,

Table 5: Comparing F-test and cross validation results for DØ measurement with different numbers of parameters.

(a) $|y| < 0.4$

F-test

|   | 3 | 4 | 5 |
|---|---|---|---|
| 4 | 1 | | |
| 5 | 1 | 0.9933 | |
| 6 | 1 | 0.9726 | 0.2181 |

Cross validation

|   | 3 | 4 | 5 |
|---|---|---|---|
| 4 | 1 | | |
| 5 | 1 | 0.9003 | |
| 6 | 1 | 0.9011 | 0.5545 |

(b) $0.4 < |y| < 0.8$

|   | 3 | 4 | 5 |
|---|---|---|---|
| 4 | 1 | | |
| 5 | 1 | 0.9987 | |
| 6 | 1 | 0.9998 | 0.9918 |

|   | 3 | 4 | 5 |
|---|---|---|---|
| 4 | 1 | | |
| 5 | 1 | 0.9661 | |
| 6 | 1 | 0.9840 | 0.8662 |

(c) $0.8 < |y| < 1.2$

|   | 3 | 4 | 5 |
|---|---|---|---|
| 4 | 1 | | |
| 5 | 1 | 0.9956 | |
| 6 | 1 | 0.9947 | 0.8929 |

|   | 3 | 4 | 5 |
|---|---|---|---|
| 4 | 1 | | |
| 5 | 1 | 0.9119 | |
| 6 | 1 | 0.9336 | 0.7427 |

(d) $1.2 < |y| < 1.6$

|   | 3 | 4 | 5 |
|---|---|---|---|
| 4 | 1 | | |
| 5 | 1 | 0.9801 | |
| 6 | 1 | 0.9748 | 0.8530 |

|   | 3 | 4 | 5 |
|---|---|---|---|
| 4 | 0.9995 | | |
| 5 | 0.9997 | 0.8693 | |
| 6 | 1 | 0.9029 | 0.7256 |

(e) $1.6 < |y| < 2.0$

|   | 3 | 4 | 5 |
|---|---|---|---|
| 4 | 1 | | |
| 5 | 1 | 0.9993 | |
| 6 | 1 | 0.9969 | 0.6202 |

|   | 3 | 4 | 5 |
|---|---|---|---|
| 4 | 0.9996 | | |
| 5 | 0.9997 | 0.8314 | |
| 6 | 0.9997 | 0.8404 | 0.5769 |

(f) $2.0 < |y| < 2.4$

|   | 3 | 4 | 5 |
|---|---|---|---|
| 4 | 1 | | |
| 5 | 0.9998 | 0.6302 | |
| 6 | 0.9996 | 0.7857 | 0.8584 |

|   | 3 | 4 | 5 |
|---|---|---|---|
| 4 | 0.9910 | | |
| 5 | 0.9917 | 0.5983 | |
| 6 | 0.9913 | 0.6889 | 0.6622 |

i.e. possible tensions in QCD interpretation would rather be related to missing physics effects on the global shape of the inclusive jet spectrum (either in experimental data or in theoretical predictions) or to issues of smoothness in the theoretical predictions. For the CMS measurement, instead, the more generous uncertainties will cover effects not explicitly included (if any), but at the cost of reducing the potential impact of the CMS data in the global PDF fits.

## 4  Summary & Conclusions

We have presented a simple method to test the smoothness of binned differential distributions, including bin-to-bin correlations, and applied it to four measurements of inclusive jet cross section in proton-antiproton or proton-proton collisions. These tests help assess the quality of a spectrum independently of and prior to a QCD interpretation.

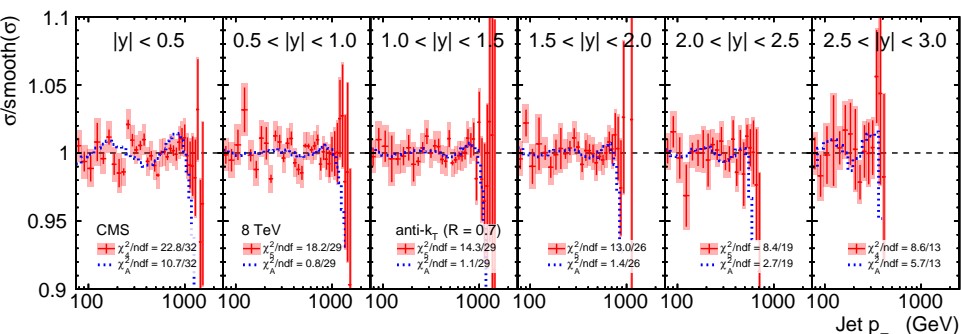

(a) The red points show the ratio to the smooth fit with the expansion of polynomials; the blue dotted line shows the ratio of Asimov data fitted with a polynomial of the same degree as for the real data in the same $y$ bin. The vertical bars on the red points show the statistical contribution only, while the shaded areas show the statistical and systematic uncertainties related to the limited statistics of the numerical analysis added in quadrature. In the $y$ axis title, $\sigma$ (smooth($\sigma$)) stands for cross section (smooth fit of the cross section).

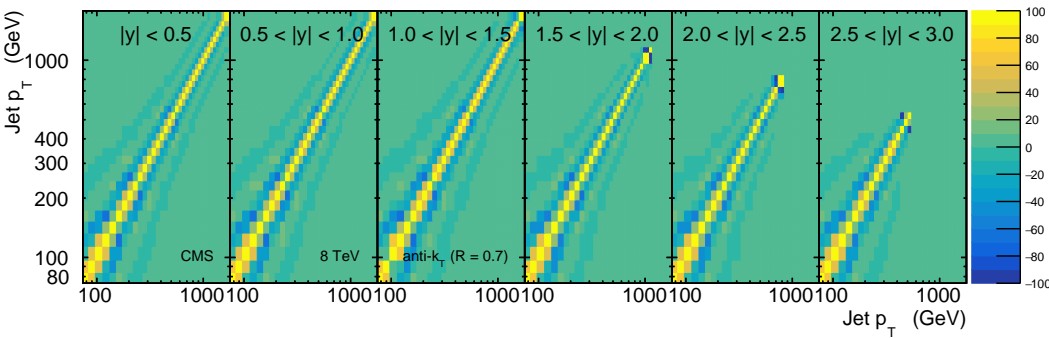

(b) Bin-to-bin correlations in percents.

Figure 6: Tests of smoothness in each $y$ bin of the CMS measurement [26].

Table 6: Fit performance in each $y$ bin of the CMS measurement [26].

(a) $\chi_n^2/\mathrm{ndf} \pm \sqrt{2\mathrm{ndf}}$

| $n$ | $|y| < 0.5$ | $0.5 < |y| < 1.0$ | $1.0 < |y| < 1.5$ | $1.5 < |y| < 2.0$ | $2.0 < |y| < 2.5$ | $2.5 < |y| < 3.0$ |
|---|---|---|---|---|---|---|
| 1 | $813.05 \pm 0.24$ | $1027.01 \pm 0.25$ | $430.17 \pm 0.25$ | $962.41 \pm 0.26$ | $934.92 \pm 0.29$ | $982.00 \pm 0.35$ |
| 2 | $46.73 \pm 0.24$ | $67.97 \pm 0.25$ | $115.66 \pm 0.25$ | $187.30 \pm 0.26$ | $190.05 \pm 0.30$ | $69.44 \pm 0.37$ |
| 3 | $6.50 \pm 0.25$ | $6.97 \pm 0.25$ | $13.09 \pm 0.25$ | $20.57 \pm 0.27$ | $16.99 \pm 0.31$ | $3.35 \pm 0.38$ |
| 4 | $0.71 \pm 0.25$ | $0.90 \pm 0.26$ | $1.04 \pm 0.26$ | $1.04 \pm 0.27$ | $0.79 \pm 0.32$ | $0.66 \pm 0.39$ |
| 5 | $0.65 \pm 0.25$ | $0.63 \pm 0.26$ | $0.49 \pm 0.26$ | $0.50 \pm 0.28$ | $0.44 \pm 0.32$ | $0.34 \pm 0.41$ |
| 6 | $0.66 \pm 0.26$ | $0.65 \pm 0.27$ | $0.45 \pm 0.27$ | $0.33 \pm 0.28$ | $0.32 \pm 0.33$ | $0.33 \pm 0.43$ |
| 7 | $0.68 \pm 0.26$ | $0.67 \pm 0.27$ | $0.45 \pm 0.27$ | $0.35 \pm 0.29$ | $0.34 \pm 0.34$ | $0.34 \pm 0.45$ |

(b) Fit probabilities

| $n$ | $|y| < 0.5$ | $0.5 < |y| < 1.0$ | $1.0 < |y| < 1.5$ | $1.5 < |y| < 2.0$ | $2.0 < |y| < 2.5$ | $2.5 < |y| < 3.0$ |
|---|---|---|---|---|---|---|
| 4 | 0.89 | 0.62 | 0.40 | 0.41 | 0.73 | 0.80 |
| 5 | 0.94 | 0.94 | 0.99 | 0.98 | 0.98 | 0.98 |
| 6 | 0.92 | 0.92 | 0.99 | 1.00 | 1.00 | 0.98 |
| 7 | 0.90 | 0.90 | 0.99 | 1.00 | 0.99 | 0.97 |

The tests of smoothness are general and may be applied to other observables, such as the dijet mass cross section, or to any measurement beyond jet physics where a smooth behaviour is expected. A tool to perform such tests of smoothness, called STEP, is provided, as well as all examples provided in this article.

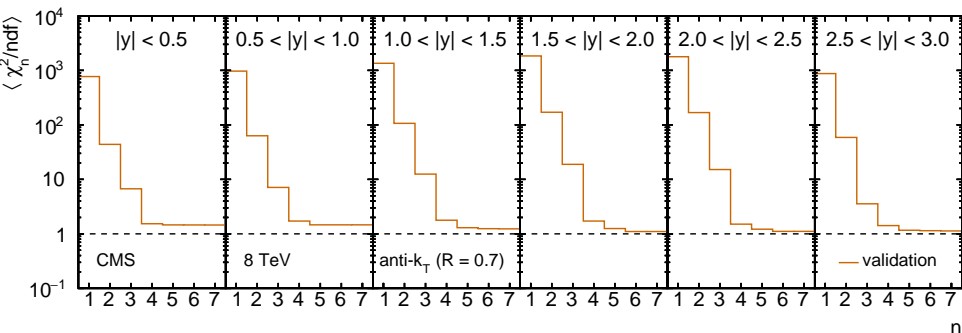

Figure 7: Average $\chi^2_n/\text{ndf}$ from validation replicas and fits of validation replicas in each $y$ bin of the CMS measurement [26].

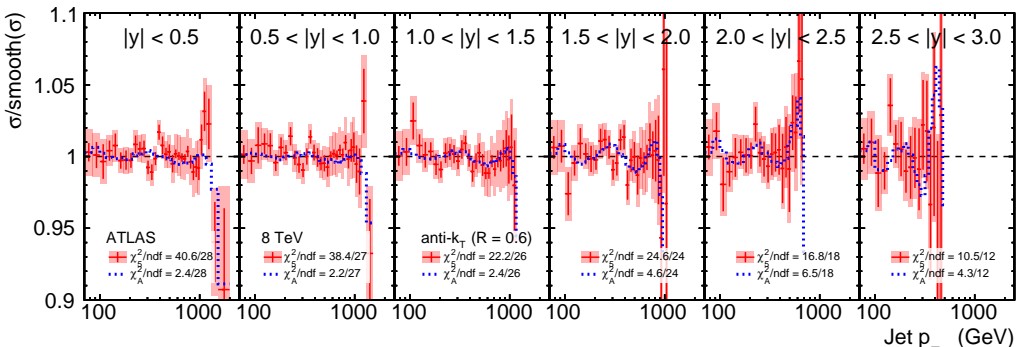

(a) The red points show the ratio to the smooth fit with the expansion of polynomials; the blue dotted line shows the ratio of Asimov data fitted with a polynomial of the same degree as for the real data in the same $y$ bin. The vertical bars on the red points show the statistical contribution only, while the shaded areas show the statistical and systematic uncertainties related to the limited statistics of the numerical analysis added in quadrature. In the $y$ axis title, $\sigma$ (smooth($\sigma$)) stands for cross section (smooth fit of the cross section).

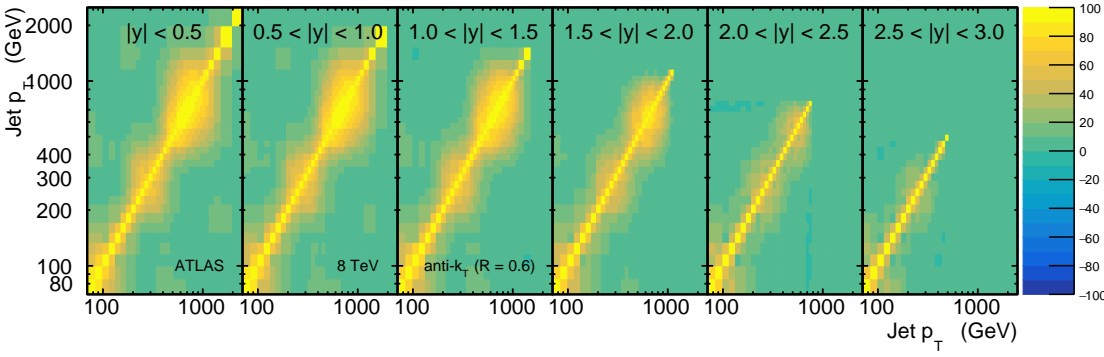

(b) Bin-to-bin correlations in percents.

Figure 8: Tests of smoothness in each $y$ bin of ATLAS measurement [27].

# Acknowledgement

This work was supported by University Hamburg, HamburgX grant LFF-HHX-03 to the Center for Data and Computing in Natural Sciences (CDCS) from the Hamburg Ministry of Science, Research, Equalities and Districts, and by the BMBF under contracts U4606BMB1901 and U4606BMB2101.

We also thank our colleagues Bogdan Malaescu, Tancredi Carli, and Zdeněk Hubáček for the interesting exchanges.

Table 7: Comparing F-test and cross validation results for CMS measurement with different numbers of parameters.

(a) $|y| < 0.5$

F-test

| | 4 | 5 | 6 |
|---|---|---|---|
| 5 | 0.9522 | | |
| 6 | 0.8698 | 0.3628 | |
| 7 | 0.7553 | 0.1686 | 0.2997 |

Cross validation

| | 4 | 5 | 6 |
|---|---|---|---|
| 5 | 0.7920 | | |
| 6 | 0.8011 | 0.5768 | |
| 7 | 0.8062 | 0.6031 | 0.5654 |

(b) $0.5 < |y| < 1.0$

| | 4 | 5 | 6 |
|---|---|---|---|
| 5 | 0.9993 | | |
| 6 | 0.9964 | 0 | |
| 7 | 0.9885 | 0.0035 | 0.0664 |

| | 4 | 5 | 6 |
|---|---|---|---|
| 5 | 0.9358 | | |
| 6 | 0.9358 | 0.4940 | |
| 7 | 0.9349 | 0.5146 | 0.5145 |

(c) $1.0 < |y| < 1.5$

| | 4 | 5 | 6 |
|---|---|---|---|
| 5 | 1 | | |
| 6 | 1 | 0.9438 | |
| 7 | 1 | 0.8833 | 0.5987 |

| | 4 | 5 | 6 |
|---|---|---|---|
| 5 | 0.9816 | | |
| 6 | 0.9851 | 0.7528 | |
| 7 | 0.9858 | 0.7727 | 0.6131 |

(d) $1.5 < |y| < 2.0$

| | 4 | 5 | 6 |
|---|---|---|---|
| 5 | 1 | | |
| 6 | 1 | 0.9991 | |
| 7 | 1 | 0.9953 | 0.0133 |

| | 4 | 5 | 6 |
|---|---|---|---|
| 5 | 0.9726 | | |
| 6 | 0.9878 | 0.8614 | |
| 7 | 0.9877 | 0.8616 | 0.5091 |

(e) $2.0 < |y| < 2.5$

| | 4 | 5 | 6 |
|---|---|---|---|
| 5 | 0.9993 | | |
| 6 | 0.9999 | 0.9894 | |
| 7 | 0.9994 | 0.9582 | 0.0740 |

| | 4 | 5 | 6 |
|---|---|---|---|
| 5 | 0.9145 | | |
| 6 | 0.9410 | 0.7917 | |
| 7 | 0.9414 | 0.7910 | 0.5127 |

(f) $2.5 < |y| < 3.0$

| | 4 | 5 | 6 |
|---|---|---|---|
| 5 | 0.9968 | | |
| 6 | 0.9910 | 0.6993 | |
| 7 | 0.9778 | 0.5615 | 0.5612 |

| | 4 | 5 | 6 |
|---|---|---|---|
| 5 | 0.8573 | | |
| 6 | 0.8645 | 0.6232 | |
| 7 | 0.8714 | 0.6460 | 0.5835 |

Table 8: Fit performance in each $y$ bin of the ATLAS measurement [27].

(a) $\chi_n^2/\text{ndf} \pm \sqrt{2\text{ndf}}$

| $n$ | $|y| < 0.5$ | $0.5 < |y| < 1.0$ | $1.0 < |y| < 1.5$ | $1.5 < |y| < 2.0$ | $2.0 < |y| < 2.5$ | $2.5 < |y| < 3.0$ |
|---|---|---|---|---|---|---|
| 1 | $513.84 \pm 0.25$ | $424.00 \pm 0.25$ | $568.23 \pm 0.26$ | $589.03 \pm 0.27$ | $634.22 \pm 0.30$ | $457.83 \pm 0.35$ |
| 2 | $18.36 \pm 0.25$ | $26.47 \pm 0.26$ | $55.22 \pm 0.26$ | $82.56 \pm 0.27$ | $111.67 \pm 0.31$ | $71.58 \pm 0.37$ |
| 3 | $6.16 \pm 0.26$ | $6.86 \pm 0.26$ | $12.36 \pm 0.27$ | $13.57 \pm 0.28$ | $13.88 \pm 0.32$ | $12.57 \pm 0.38$ |
| 4 | $1.75 \pm 0.26$ | $2.71 \pm 0.27$ | $2.88 \pm 0.27$ | $2.65 \pm 0.28$ | $2.17 \pm 0.32$ | $2.20 \pm 0.39$ |
| 5 | $1.45 \pm 0.27$ | $1.42 \pm 0.27$ | $0.85 \pm 0.28$ | $1.03 \pm 0.29$ | $0.93 \pm 0.33$ | $0.88 \pm 0.41$ |
| 6 | $1.47 \pm 0.27$ | $1.43 \pm 0.28$ | $0.87 \pm 0.28$ | $1.00 \pm 0.29$ | $0.69 \pm 0.34$ | $0.93 \pm 0.43$ |
| 7 | $1.43 \pm 0.28$ | $1.48 \pm 0.28$ | $0.72 \pm 0.29$ | $1.04 \pm 0.30$ | $0.73 \pm 0.35$ | $0.74 \pm 0.45$ |

(b) Fit probabilities

| $n$ | $|y| < 0.5$ | $0.5 < |y| < 1.0$ | $1.0 < |y| < 1.5$ | $1.5 < |y| < 2.0$ | $2.0 < |y| < 2.5$ | $2.5 < |y| < 3.0$ |
|---|---|---|---|---|---|---|
| 4 | 0.01 | 0.00 | 0.00 | 0.00 | 0.00 | 0.01 |
| 5 | 0.06 | 0.07 | 0.68 | 0.43 | 0.54 | 0.57 |
| 6 | 0.05 | 0.07 | 0.64 | 0.46 | 0.82 | 0.51 |
| 7 | 0.07 | 0.06 | 0.83 | 0.40 | 0.76 | 0.68 |

Table 9: Comparing F-test and cross validation results for ATLAS measurement with different numbers of parameters.

(a) $|y| < 0.5$

F-test

| | 4 | 5 | 6 |
|---|---|---|---|
| 5 | 0.9869 | | |
| 6 | 0.9638 | 0.555 | |
| 7 | 0.9583 | 0.6749 | 0.8002 |

Cross validation

| | 4 | 5 | 6 |
|---|---|---|---|
| 5 | 0.9427 | | |
| 6 | 0.9515 | 0.6851 | |
| 7 | 0.9707 | 0.8244 | 0.7868 |

(b) $0.5 < |y| < 1.0$

F-test

| | 4 | 5 | 6 |
|---|---|---|---|
| 5 | 1 | | |
| 6 | 0.9999 | 0.6467 | |
| 7 | 0.9996 | 0.3746 | 0.2386 |

Cross validation

| | 4 | 5 | 6 |
|---|---|---|---|
| 5 | 0.9985 | | |
| 6 | 0.9989 | 0.7096 | |
| 7 | 0.9987 | 0.7217 | 0.5752 |

(c) $1.0 < |y| < 1.5$

F-test

| | 4 | 5 | 6 |
|---|---|---|---|
| 5 | 1 | | |
| 6 | 1 | 0.4720 | |
| 7 | 1 | 0.9490 | 0.9804 |

Cross validation

| | 4 | 5 | 6 |
|---|---|---|---|
| 5 | 1 | | |
| 6 | 1 | 0.6189 | |
| 7 | 1 | 0.8693 | 0.8583 |

(d) $1.5 < |y| < 2.0$

F-test

| | 4 | 5 | 6 |
|---|---|---|---|
| 5 | 1 | | |
| 6 | 1 | 0.7827 | |
| 7 | 1 | 0.5337 | 0.1527 |

Cross validation

| | 4 | 5 | 6 |
|---|---|---|---|
| 5 | 0.9994 | | |
| 6 | 0.9995 | 0.7357 | |
| 7 | 0.9997 | 0.7388 | 0.5417 |

(e) $2.0 < |y| < 2.5$

F-test

| | 4 | 5 | 6 |
|---|---|---|---|
| 5 | 0.9999 | | |
| 6 | 1 | 0.9853 | |
| 7 | 0.9999 | 0.9439 | 0.0021 |

Cross validation

| | 4 | 5 | 6 |
|---|---|---|---|
| 5 | 0.994 | | |
| 6 | 0.9973 | 0.8712 | |
| 7 | 0.9973 | 0.8711 | 0.5056 |

(f) $2.5 < |y| < 3.0$

F-test

| | 4 | 5 | 6 |
|---|---|---|---|
| 5 | 0.9993 | | |
| 6 | 0.9965 | 0.4138 | |
| 7 | 0.9971 | 0.8238 | 0.9187 |

Cross validation

| | 4 | 5 | 6 |
|---|---|---|---|
| 5 | 0.9841 | | |
| 6 | 0.9843 | 0.5951 | |
| 7 | 0.9901 | 0.8122 | 0.8055 |

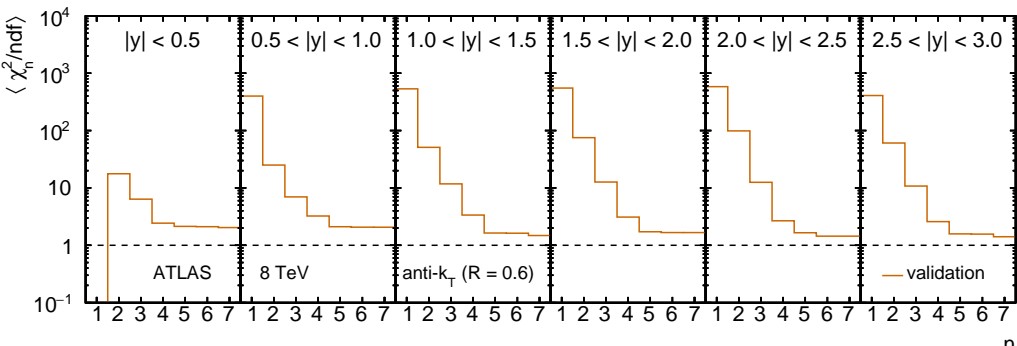

Figure 9: Average $\chi_n^2/\mathrm{ndf}$ from validation replicas and fits of validation replicas in each $y$ bin of the ATLAS measurement [27].

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
