# Peer review of "Step: a tool to perform tests of smoothness on differential distributions based on expansion of polynomials"

_SciPost Physics Core, doi:SciPost Phys. Core 6, 040 (2023)_

## Round 2 · Referee Report · Anonymous (Referee 1) · 2022-1-7

Strengths

1 The underlying idea is very simple

Weaknesses

1 The motivation for the paper is questionable
2 The details of the methodology are unclear
3 The conclusion and examples are unclear
4 The results are quite modest

Report

The paper presents an algorithm for fitting spectra with Chebyshev polynomials, that is presented as a "test of smoothness". The main motivation for this appears to have to do with PDF fits.

First of all, it should be mentioned that the motivation is quite weak: firstly, available evidence suggests that the problems in PDF fits the Authors refer to have to do with properties of the experimental covariance matrix (specifically its being ill-conditioned) so it is unclear that smoothness has anything to do with them. Moreover, given that PDFs are related to data through a convolution it is unclear that data smoothness could possibly be a problem.

Coming to the method itself, what it amounts to is simply an iterative fit of Chebyshev polynomials. This is a very simple idea and it is unclear that having implemented it in a piece of code is worthy of publication.

The motivations for the use of Chebyshev polynomials appear to be very generic: on pag. 2 the authors list a couple desirable properties of these polynomials, but it is unclear that other orthogonal polynomials wouldn't have similar or perhaps better properties. So the specific choice appears to be somewhat haphazard.

Furthermore, there are many specific points that make both the details of the methodology and its proposed uses somewhat obscure:

1) at point 3 on pag. 3 as a stopping criterion it is mentioned that $\chi^2$ is "compatible with the number of degrees of freedom". What does "compatible" mean? What is the exact criterion? 2) on pag. 5 a "procedure of early stopping" is mentioned. But this was never mentioned previously! What is this early stopping? 3) The authors talk about a "test" but what they present is simply a sequence of fits. Presumably, the test would amount to comparing the fit results to some criterion? It is completely unclear what the test is.

Finally, the discussion of results is somewhat disappointing: the authors end up discussing qualitative features of the spectra that they display, but it is not at all clear what the added value of their proposed "test" is.

In summary, I do not think that this paper contains enough original material or results to justify a publication in a scientific journal.

  • validity: ok
  • significance: low
  • originality: low
  • clarity: low
  • formatting: reasonable
  • grammar: good

Author:  Patrick Connor  on 2022-06-13  [id 2577]

(in reply to Report 1 on 2022-01-07)

Dear referee,

thank you very much for reviewing our paper. Please find answers point by point herebelow.

The paper has undergone significant changes since your first review.

Best regards, Patrick Connor & Radek Zlebcik

First of all, it should be mentioned that the motivation is quite weak: firstly, available evidence suggests that the problems in PDF fits the Authors refer to have to do with properties of the experimental covariance matrix (specifically its being ill-conditioned) so it is unclear that smoothness has anything to do with them. Moreover, given that PDFs are related to data through a convolution it is unclear that data smoothness could possibly be a problem.

The smoothness of the distribution is certainly not a sufficient condition for the usability of the data in a QCD interpretation. However, it is a necessary condition. Indeed, by construction of the QCD fits, where the PDFs at a starting scale are first evolved and consequently convoluted with the partonic cross section, the resulting spectra must be smooth, as both operations smear the orginal PDF. In the QCD fits, steps or outliers in the experimental spectrum that are not covered by the statistical uncertainties will lead to larger 𝜒2/ndf values for the data set, sometimes even prohibiting the use of the experimental data at all.

In the new version of the paper, we now demonstrate the smoothness of the predictions by repeating our test of smoothness on Asimov data, which have, by construction, central values identical with the theoretical model. As expected, the fit performance of Asimov data is at least as good as and often significantly better than the that of the real data.

We have now also clarified what we mean with covariance matrix, which describes the statistical and systematic correlations among the bins of the truth level distribution, to make sure that it is not confused with the response matrix, which is used in the unfolding procedure to describe the migrations from the truth level to the detector level. The response matrix, indeed known to often be ill-conditioned, is sometimes called correlation matrix, which can lead to the confusion.

Coming to the method itself, what it amounts to is simply an iterative fit of Chebyshev polynomials. This is a very simple idea and it is unclear that having implemented it in a piece of code is worthy of publication.

We believe that such tool is useful for experimentalists to test the quality of their data distributions before their release. In fact, since our first submission, colleagues from the ATLAS Collaboration have already started to use our technique to investigate their data (Ref. [18] in the new version of the paper draft).

In the context of inclusive jet measurement, it is an alternative to tests of smoothness using QCD fits, without the danger of biasing toward the theory. In the paper draft, we have shown that it can help to identify steps corresponding to trigger thresholds. Another application consists in testing the procedure of unfolding, as a similar fit performance is expected before and after the procedure (although the fitted smooth function is always different). For instance, a typical mistake performed in the procedure of unfolding is to underestimate the uncertainties from the simulated data, used for the response matrix and for the estimation of migrations through the edges of the phase space. Unfortunately, this cannot be illustrated directly since only unfolded (i.e. truth level) data are public. On the other hand, certain uncertainties may be overestimated, as is probably the case for the flat 1% bin-to-bin uncorrelated uncertainty added in the 8-TeV inclusive measurement from the CMS Collaboration; if experimentalists had had such a test available at that time, they may have been able to better identify the origin of the deviations from a smooth behaviour before the unfolding (e.g. steps due to triggers, as illustrated in the paper draft), without any need to increase the experimental uncertainty. It would lead to higher impact of the data in the global PDF fits.

We have significantly modified Section 3, trying to clarify all these points.

The motivations for the use of Chebyshev polynomials appear to be very generic: on pag. 2 the authors list a couple desirable properties of these polynomials, but it is unclear that other orthogonal polynomials wouldn’t have similar or perhaps better properties. So the specific choice appears to be somewhat haphazard.

We have also tested alternative orthogonal bases, such as Legendre polynomials, which seem to work too; instead, the use of standard polynomials (1, 𝑋, 𝑋^2, etc.) results in limited stability of the iterative procedure. We have now mentioned in the paper draft the possibility to use alternative bases, and adapted the C++ implementation to allow the user to use alternative polynomial bases.

However, the aim of this paper is not to find all polynomial bases that work. For the purpose of our test, which is to find a function allowing to factor out the global shape of the spectrum and to focus on the description of the scattering of the data points around a smooth behaviour, it is sufficient to find only one family of polynomials.

Furthermore, there are many specific points that make both the details of the methodology and its proposed uses somewhat obscure: 1. at point 3 on pag. 3 as a stopping criterion it is mentioned that χ2 is “compatible with the number of degrees of freedom”. What does “compatible” mean? What is the exact criterion? 2. on pag. 5 a “procedure of early stopping” is mentioned. But this was never mentioned previously! What is this early stopping? 3. The authors talk about a “test” but what they present is simply a sequence of fits. Presumably, the test would amount to comparing the fit results to some criterion? It is completely unclear what the test is.

  1. A 𝜒2 distribution is expected to be centred at the number of degrees of freedom and its variance is expected to be twice the number of degrees of freedom (𝑘). With “compatible”, we mean that we expect the following inequality to be satisfied: |𝜒2−𝑘|<√2𝑘. This is now clarified in the text.
  2. The procedure of (early) stopping is explicitly described in the definition of the fit algorithm. Since the former version of the paper draft, we have implemented two more early stopping criteria, one based on F-test, and another based on cross validation. We have added detailed descriptions and discussions on the outputs of the different early stopping criteria.
  3. We “test” whether the scattering of the points around a smooth function is described by the uncertainties (typically the statistical uncertainties). This smooth function includes one hyperparameter describing the number of fit parameters. A bad fit performance (e.g. 𝜒2/𝑘≈4) for a large value of this hyperparameter would indicate that the uncertainties alone cannot cover the deviations. For instance, in the absence of bin-to-bin partial correlations (e.g. D0, CDF), this would typically appear as outliers or steps. In the present of such correlations (e.g. CMS, ATLAS), introduced for instance through the procedure of unfolding, a direct interpretation in terms of outliers or steps is no longer guaranteed, but can still indicate whether the provided uncertainties are sufficient.

We have significantly changed the text to clarify these points.

Finally, the discussion of results is somewhat disappointing: the authors end up discussing qualitative features of the spectra that they display, but it is not at all clear what the added value of their proposed “test” is.

The approach followed by experimentalists to test the quality of their data is to perform a QCD interpretation with software such as xFitter. However, this assumes that the theory reasonably describes the data within certain systematic uncertainties. Large 𝜒2 values are then difficult to interpret, since they may have various origins. With our approach, we factor out possible sources of tensions related to underestimated uncorrelated uncertainties and to possible issues inherent to the theory curves. The test of smoothness focuses on the quality of the data at any stage of the experimental analysis. The smoothness is a very weak—hence powerful—assumption.

Publishing an experimental analysis is a long process that easily spans over a few years within a collaboration. Once the data have been published, it is generally too late to investigate issues in the experimental analysis. By publishing our method, we want to provide an additional tool for our present and future colleagues to test the quality of their spectra before publishing them. The impact of such a test will lead to an improved quality of the published data and likely to a better perturbative QCD interpretation (under the assumption that QCD is the right theory).

Anonymous on 2022-12-09  [id 3118]

(in reply to Patrick Connor on 2022-06-13 [id 2577])

I am satisfied with the Author's revision and replies to my original criticism. My only residual comment is that I find the terminology "early stopping" very confusing. To me, "early stopping" suggests that there exists also "late stopping" - and possibly "correct stopping", which is neither early nor late? Indeed, before the Author's explanations, I misunderstood what they meant here. While in actual fact, what the Authors call "early stopping" is just the way they implement their test. Also, in this respect I find a bit confusing that the various criteria that they suggest correspond to rather different ways of implementing a smoothness test: e.g. with criterion 3 (the process is interrupted when a given p value is reached) surely the test is passed or failed according to whether the fit does or does not stop, while with criterion 4 (chi^2 stops improving) the fit is passed or not according to what the final chi2 value is. Upon careful reading of the paper this may be understood, but it seems to me that the paper would be easier to understand if this were spelled out explicitly.

---

## Round 2 · Referee Report · Anonymous (Referee 2) · 2022-2-7

Strengths

1) The idea of examining the smoothness of a differential distribution of jet related observables, such that they can be easily incorporated in PDF fits etc, is certainly a very useful one. Such tests have been extensively performed by the ATLAS and CMS experiments recently, and DO and CDF previously, especially in the context of searches of narrow (and wide) resonances decaying to jets, see for example a review :

R. M. Harris and K. Kousouris, “Searches for dijet resonances at hadron colliders”, Int. J. Mod. Phys. A 26 (2011) 5005, doi:10.1142/S0217751X11054905, arXiv:1110.5302.

2) The idea to provide a tool to be able to perform such tests quickly and in an automated way is certainly commendable, and should be pursued and developed further with the addition of more functional forms and tests to determine the needed number of freely floating parameters.

Weaknesses

Specific comments and questions on the current paper draft

1) Have you examined other families of functions, as for example in the review above (eq. 47, 48, 49) , to perform the same tests? If not, and there is a specific reason why only Chebyshev polynomials are utilized, please state this in the paper.

2) Please provide the probabilities of fits, rather than only chisquare/ndf, in order to be able to assess how likely these descriptions of the data are.

3) Have you tried utilizing the Fisher Test to determine how many parameters to use in each function, and when to stop? If not, and given that this test is widely used in searches to test how many parameters to use in the functions describing the background, please consider it, and state if what you do is equivalent and/or superior.

Fisher, R. A. (1922). "On the interpretation of χ2 from contingency tables, and the calculation of P". Journal of the Royal Statistical Society. 85 (1): 87–94. doi:10.2307/2340521. JSTOR 2340521.

4) How do you address the possible inadequacy of the function you utilize to describe the experimental data? In searches often the pulls (data-fit/data statistical uncertainty) of the distributions/fits are also examined to see if there are continuous runnings that would indicate that a different function or more parameters should be added. See for example Fig.5 of

CMS Collaboration ,"Search for high mass dijet resonances with a new background prediction method in proton-proton collisions at $\sqrt{s} =$ 13 TeV", JHEP 05 (2020), 033

and Fig.1 of

ATLAS Collaboration, “Search for new phenomena in dijet events using 37 fb −1 of pp √ collision data collected at s =13 TeV with the ATLAS detector” ,Phys. Rev. D96 (2017) 052004, doi:10.1103/PhysRevD.96.052004, arXiv:1703.09127.

Report

If the points above are addressed I would recommend publishing in this journal.

Requested changes

The points above listed as "Weaknesses" would be good to be addressed and the relevant and needed information to be incorporated in the paper.

  • validity: high
  • significance: high
  • originality: high
  • clarity: good
  • formatting: good
  • grammar: good

Author:  Patrick Connor  on 2022-06-13  [id 2578]

(in reply to Report 2 on 2022-02-07)

Dear referee,

thank you very much for your comments and suggestions. Please find answers to the points that you have raised below our signature.

Since your first review, the paper has been applied a significant amount of modifications and improvements.

Best regards, Radek Zlebcik & Patrick Connor

R. M. Harris and K. Kousouris, “Searches for dijet resonances at hadron colliders”, Int. J. Mod. Phys. A 26 (2011) 5005, doi:10.1142/S0217751X11054905, arXiv:1110.5302.

1) Have you examined other families of functions, as for example in the review above (eq. 47, 48, 49) , to perform the same tests? If not, and there is a specific reason why only Chebyshev polynomials are utilized, please state this in the paper.

We have tested the other families of functions from the review that you have mentioned. We have added them to the paper draft for all four measurements (the green dotted points labelled "HK" in the legend). In general, for these two measurements, they give comparable description with similar number of parameters. However, for example, for the ATLAS data points which have smaller uncertainties, these functions do not describe shape of data well and more general perscription with more parameters is needed. Here, advantage of our approach is that the generalisation of the model is straighforward.

We have tested also other orthogonal polynomials, like Legendre polynomials: they exhibit similar performance. In our C++ implementation, we have now implemented an option to use alternative bases of polynomials in the fits.

2) Please provide the probabilities of fits, rather than only χ2/ndf, in order to be able to assess how likely these descriptions of the data are.

We have added tables with fit probabilities in the new version.

3) Have you tried utilizing the Fisher Test to determine how many parameters to use in each function, and when to stop? If not, and given that this test is widely used in searches to test how many parameters to use in the functions describing the background, please consider it, and state if what you do is equivalent and/or superior.

Fisher, R. A. (1922). "On the interpretation of χ2 from contingency tables, and the calculation of P". Journal of the Royal Statistical Society. 85 (1): 87–94. doi:10.2307/2340521. JSTOR 2340521.

We have added the F-test as an alternative early stopping criterion. Furthermore, we have also implemented an early stopping criterion based on statistical replicas of the data sets and cross validation. We have added tables and detailed discussions on the results of the different early stopping criteria.

4) How do you address the possible inadequacy of the function you utilize to describe the experimental data? In searches often the pulls (data-fit/data statistical uncertainty) of the distributions/fits are also examined to see if there are continuous runnings that would indicate that a different function or more parameters should be added. See for example Fig.5 of

CMS Collaboration ,"Search for high mass dijet resonances with a new background prediction method in proton-proton collisions at √s=13 TeV", JHEP 05 (2020), 033

and Fig.1 of

ATLAS Collaboration, “Search for new phenomena in dijet events using 37 fb −1 of pp √ collision data collected at s =13 TeV with the ATLAS detector” ,Phys. Rev. D96 (2017) 052004, doi:10.1103/PhysRevD.96.052004, arXiv:1703.09127.

The functionality of obtaining the pulls was already provided in the GitLab repository of the tool, but not detailed in the first version of the paper. We have now added figures in the paper and compared the fit performance with our approach. In this way, one ensures that the function really describes overall shape and is suitable to spot the outliers.

---

## Round 3 · Referee Report · Anonymous (Referee 3) · 2022-9-1

Strengths

1) These paper presents a new, complementary to the existing ones, approach of i) assessing the smoothness of QCD differential distributions and the appropriateness of bin-by-bin uncertainties together with their correlations, and ii) performing a high-quality smooth fits using empirical parametrizations and an iterative method, determining the number of needed free parameters using several different statistical methods.

2) This approach is automated, requires less tuning of initial parameter and parameter ranges, and is easily accessible by both the experimental Collaborations, and the phenomenology and theory community.

3) To validate this approach and show its performance several tests have been performed and are presented in this paper, using both Tevatron and LHC data, showing how successful the method is in describing differential jet pT distributions, and in assessing whether the uncertainties accompanying them are conservative or optimistic. Also, comparisons with existing methods are also presented showing complementary and in some cases advantageous results.

Report

These paper presents a new, complementary to the existing ones, approach of i) assessing the smoothness of QCD differential distributions and the appropriateness of bin-by-bin uncertainties together with their correlations, and ii) performing a high-quality smooth fits using empirical parametrizations and an iterative method, determining the number of needed free parameters using several different statistical methods.

This approach is automated, requires less tuning of initial parameter and parameter ranges, and is easily accessible by both the experimental Collaborations, and the phenomenology and theory community.

To validate this approach and show its performance several tests have been performed and are presented in this paper, using both Tevatron and LHC data, showing how successful the method is in describing differential jet pT distributions, and in assessing whether the uncertainties accompanying them are conservative or optimistic. Also, comparisons with existing methods are also presented showing complementary and in some cases advantageous results.

Requested changes

1)Please define in the text what the Y axis on Figures 2,4,6,8, by clarifying what sigma and smooth(sigma) are.
2) If possible, please improve the markers on Figures 2a,4a, 6a, 8a, using, for example, different colours for the statistical and systematic components.

---

## Round 3 · Author Response

Dear Editor,

thank you for your invitation to resubmit.

We have carefully considered all the comments from the referees and applied changes to the paper draft accordingly. In addition, we have applied changes according to feedback on the pre-print from the community.

Best regards,
Patrick Connor & Radek Žlebčík

---

## Round 3 · List of Changes

- The technique has been checked successfully with alternative bases, e.g. Legendre polynomials.
- Two additional early-stopping criteria have been implemented, namely F-test and cross validation on replicas.
- The fit performance of the Step algorithm has been comparsed with that of an alternative function suggested by one of the referees.
- A comparison of the fit performance with Asimov data sets to show expectation has been added.
- Additional uncertainties on the ATLAS data coming from the in-situ calibration have been included in the fit.
- Distributions of pulls have been included for the Tevatron measurements.
- Fit probability tables are now shown.
- Global chi2/ndf values for the LHC measurement considering also bin-to-bin correlations have been provided.
- In general, we have significantly extended the discussions.
- The figure with the comparison of the chi2/ndf per rapidity bin and per experiment has been removed.

---

## Round 4 · List of Changes

Answerings to the 1st report:
1) We have clarified sigma and smooth(sigma) in the Y axis on Figures 2,4,6,8, in the captions.
2) We have improved the style of Figures 2a,4a, 6a, 8a, to better differentiate the statistical and systematic components.
Answering to the 2nd report:
3) We have changed "early stopping" into "stopping", and explicitly said that the four stoppings correspond to 4 flavours of the test.
Additional changes:
4) We have removed occurences of "very" throughout the text.
5) We have provided a reference to the midpoint cone algorithm used by the D0 and CDF Collaborations.

---

## Editorial Decision

published